# Thymoquinone Inhibits JAK/STAT and PI3K/Akt/ mTOR Signaling Pathways in MV4-11 and K562 Myeloid Leukemia Cells

**DOI:** 10.3390/ph15091123

**Published:** 2022-09-08

**Authors:** Futoon Abedrabbu Al-Rawashde, Abdullah Saleh Al-wajeeh, Mansoureh Nazari Vishkaei, Hanan Kamel M. Saad, Muhammad Farid Johan, Wan Rohani Wan Taib, Imilia Ismail, Hamid Ali Nagi Al-Jamal

**Affiliations:** 1School of Biomedicine, Faculty of Health Sciences, Universiti Sultan Zainal Abidin (UniSZA), Terengganu 21300, Malaysia; 2Department of Anatomy and Histology, Faculty of Medicine, Mutah University, Al-Karak 61710, Jordan; 3Anti-Doping Lab Qatar, Doha P.O. Box 27775, Qatar; 4School of Pharmacy, University of 17 August 1945, Jakarta 14350, Indonesia; 5Department of Haematology, School of Medical Sciences, Universiti Sains Malaysia, Kubang Kerian, Kelantan 16150, Malaysia

**Keywords:** thymoquinone, JAK/STAT, PI3K, Akt, mTOR, AML, CML

## Abstract

Constitutive activation of Janus tyrosine kinase-signal transducer and activator of transcription (JAK/STAT) and Phosphatidylinositol 3-kinase/Akt/mammalian target of rapamycin (PI3K/Akt/mTOR) signaling pathways plays a crucial role in the development of acute myeloid leukemia (AML) and chronic myeloid leukemia (CML). Thymoquinone (TQ), one of the main constituents of *Nigella sativa*, has shown anti-cancer activities in several cancers. However, the inhibitory effect mechanism of TQ on leukemia has not been fully understood. Therefore, this study aimed to investigate the effect of TQ on JAK/STAT and PI3K/Akt/mTOR pathways in MV4-11 AML cells and K562 CML cells. *FLT3*-ITD positive MV4-11 cells and *BCR-ABL* positive K562 cells were treated with TQ. Cytotoxicity assay was assessed using WSTs-8 kit. The expression of the target genes was evaluated using RT-qPCR. The phosphorylation status and the levels of proteins involved in JAK/STAT and PI3K/Akt/mTOR pathways were investigated using Jess western analysis. TQ induced a dose and time dependent inhibition of K562 cells proliferation. TQ significantly downregulated *PI3K*, *Akt*, and *mTOR* and upregulated *PTEN* expression with a significant inhibition of JAK/STAT and PI3K/Akt/mTOR signaling. In conclusion, TQ reduces the expression of *PI3K*, *Akt*, and *mTOR* genes and enhances the expression of *PTEN* gene at the mRNA and protein levels. TQ also inhibits JAK/STAT and PI3K/Akt/mTOR pathways, and consequently inhibits proliferation of myeloid leukemia cells, suggesting that TQ has potential anti-leukemic effects on both AML and CML cells.

## 1. Introduction

Leukemia is a group of blood cancers characterized by accumulated acquired somatic genetic alterations in the hematopoietic stem cells (HSCs) resulting in uncontrolled proliferation and differentiation [1,2]. 

Acute myeloid leukemia (AML) is characterized by distinct genetic abnormalities, such as *FLT3*-ITD mutation [3]. *FLT3*-ITD mutation is the most frequently identified genetic event in AML that results in the activation of several signaling pathways, including Janus activated kinase/signal transducer and activator of transcription (JAK/STAT) and Phosphatidylinositol 3-kinase/Akt/mammalian target of rapamycin (PI3K/Akt/mTOR) signaling pathways [4,5]. 

Chronic myeloid leukemia (CML) is initiated by a chromosomal translocation t (9; 22), which results in the formation of a Philadelphia (Ph) chromosome with the BCR-ABL fusion gene [6,7]. The resulting oncoprotein (BCR-ABL) has a constitutive kinase activity that hyperactivates many signaling pathways, including JAK/STAT and PI3K/Akt/mTOR pathways, leading to enhanced hematopoietic cells survival and proliferation [8]. 

STAT proteins are downstream targets of JAKs that are essential transcription factors in regulating several cellular processes such as proliferation, cell cycle progression, and apoptosis [9]. STATs are constitutively activated through increased production of growth factors [10]. In addition, STATs are hyperactivated by mutated upstream tyrosine kinases, including FLT3-ITD and JAK2 or BCR-ABL oncoprotein [8,11]. Constitutive activation of JAK/STAT pathway has been associated with the pathogenesis of hematological malignancies, including leukemia [12]. Activated JAK2, STAT3, and STAT5 are confirmed in variable leukemia cells [13,14]. Thus, proteins involved in the JAK/STAT pathway are important targets for the treatment of leukemia.

The PI3K/Akt/mTOR signaling pathway plays a critical role in the regulation of cell proliferation, apoptosis, and cell cycle progression [15]. Hyperactivation of the PI3K/Akt/mTOR pathway is crucial for the development of various malignancies, including leukemia [16]. The activity of PI3K is negatively regulated by the phosphatase and tensin homolog (PTEN) protein [17]. Although mutations of PTEN are uncommon in myeloid leukemia, they are often inactivated in AML and CML [18]. Therefore, inhibiting PI3K/Akt/mTOR signaling represents a significant therapeutic target for leukemia treatment.

The outcome of leukemic patient therapy remains poor, with an unsatisfactory survival rate, and chemotherapy is still the first-line treatment for leukemia [19]. The development of tyrosine kinase inhibitors (TKIs) has resulted in significant advancements in leukemia treatment [8]. However, majority of leukemia patients treated with TKIs do not achieve a complete cure and exhibit drug resistance under prolonged therapy [20]. Moreover, several adverse side effects are associated with TKIs treatment [21]. Therefore, identifying alternative therapeutic candidates derived from natural products could substantially affect leukemia treatment.

Natural products are an effective and safe treatment for cancers including leukemia [22]. Thymoquinone (TQ) is a phytochemical compound and is one of the major bioactive components of *Nigella sativa* seeds [23]. TQ has been reported to inhibit cancer cell growth in various cancers [24,25,26]. However, the effect of TQ on leukemia cells has not been fully understood. Therefore, this study aimed to investigate the effect of TQ on JAK/STAT and PI3K/Akt/mTOR pathways in MV4-11 AML cells and K562 CML cells. For this purpose, cytotoxicity analysis was performed to evaluate the effect of TQ on the proliferation of K562 cells. RT-qPCR was performed to assess the effect of TQ on the expression of *PI3K*, *Akt*, *mTOR*, and *PTEN* genes in MV4-11 and K562 cells. Jess simple western analysis was conducted to evaluate the effect of TQ on JAK/STAT and PI3K/Akt/mTOR signaling pathways.

## 2. Results

### 2.1. Thymoquinone Exhibits a Growth Inhibitory Effect in K562 Cells

The cell viability was investigated using WST-8 assay. Exposure of K562 cells to an increasing concentration of TQ (3 μM, 6 μM, 9 μM, 12 μM, 15 μM, 18 μM, 21 μM, 24 μM, 27 μM, and 30 μM) led to a concentration and time-dependent decrease in the viability of the cells, with IC_50_ values of 23 µM, 15 µM, and 11 µM cells after 24 h, 48 h, and 72 h of treatment (Figure 1). The cells showed the lowest IC_50_ value after 72 h of TQ treatment (Figure 1).

In our previous study, we found that TQ decreased the viability of MV4-11 cells in a dose and time-dependent manner with IC_50_ values of 7.8 µM, 5.5 µM, and 3.8 µM after 24 h, 48 h, and 72 h of treatment, respectively [27].

### 2.2. Thymoquinone Downregulates Several Genes in PI3K/Akt/mTOR Pathway in MV4-11 and K562 Cells

The mRNA expression levels of *PI3K*, *Akt*, *mTOR*, and *PTEN* genes in MV4-11 and K562 cells were evaluated by RT-qPCR before and after treatment with TQ. MV4-11 cells were incubated with the IC_50_ of TQ (5.5 μM) for 48 h. The results of RT-qPCR showed that TQ induced a significant decrease in the mRNA levels of PI3K by 6.3-fold (*p* < 0.001), Akt by 17.5-fold (*p* < 0.001), and mTOR by 8.2-fold (*p* < 0.001) in MV4-11 cells compared to untreated cells. On the other hand, TQ induced a slight increase in the mRNA level of PTEN in MV4-11 cells, but the increase was not significant (*p* > 0.05) compared to untreated cells (Figure 2A). 

K562 cells were treated with the IC_50_ of TQ (15 μM) for 48 h. The results of RT-qPCR showed that TQ induced a significant decrease in the mRNA levels of PI3K by 25-fold (*p* < 0.001), Akt by 10.5-fold (*p* < 0.001), and mTOR by 2.7-fold (*p* < 0.001) compared to untreated K562 cells. On the other hand, TQ increased the mRNA level of PTEN in K562 cells by 1.3-fold (*p* < 0.05) compared to untreated cells (Figure 2B). 

### 2.3. Thymoquinone Inhibits JAK/STAT Signaling in MV4-11 Cells

The influence of TQ on the protein levels and the phosphorylation status of STAT3, STAT5, and JAK2 were evaluated in MV4-11 cells using Jess simple western analysis before and after treatment with TQ. MV4-11 cells were incubated with the IC_50_ of TQ (5.5 μM) for 48 h. Jess simple western analysis results showed that TQ significantly inhibited JAK/STAT signaling in MV4-11 cells by reducing the phosphorylation of STAT3 (*p* < 0.001), STAT5 (*p* < 0.05), and JAK2 (*p* < 0.05) compared to untreated cells (Figure 3 and Table 1). The results also showed that TQ significantly reduced the protein levels of STAT3 (*p* < 0.05), STAT5 (*p* < 0.001), and JAK2 (*p* < 0.01) in MV4-11 cells compared to untreated cells (Figure 3 and Table 1).

### 2.4. Thymoquinone Inhibits PI3K/Akt/mTOR Signaling in MV4-11 Cells

TQ’s influence on the protein levels and the phosphorylation status of PI3K, Akt, and the protein expression levels of PTEN were evaluated in MV4-11 cells using Jess simple western analysis. MV4-11 cells were incubated with the IC_50_ of TQ (5.5 μM) for 48 h. The results demonstrated that TQ significantly inhibited PI3K/Akt/mTOR signaling in MV4-11 cells by decreasing the phosphorylation of p-PI3K (*p* < 0.001) and p-Akt (*p* < 0.001) compared to untreated cells (Figure 4 and Table 1). The results also demonstrated that TQ significantly reduced the protein levels of PI3K (*p* < 0.001) and Akt (*p* < 0.001) in MV4-11 cells compared to untreated cells. On the other hand, TQ induced a significant increase in the protein level of PTEN (*p* < 0.001) in MV4-11 cells compared to untreated cells (Figure 4 and Table 1). 

### 2.5. Thymoquinone Inhibits PI3K/Akt/mTOR Signaling in K562 Cells

The effects of TQ on the protein expression levels and the phosphorylation status of PI3K and Akt, and the protein expression levels of PTEN were evaluated in K562 cells using Jess simple western analysis. K562 cells were incubated with the IC_50_ of TQ (15 μM) for 48 h. There was a significant inhibition of PI3K/Akt/mTOR signaling in K562 cells by decreasing the phosphorylation of p-PI3K (*p* < 0.001) and p-Akt (*p* < 0.001) compared to untreated cells (Figure 5 and Table 2). The results also demonstrated that TQ significantly reduced the protein levels of PI3K (*p* < 0.001) and Akt (*p* < 0.001) compared to untreated cells. On the other hand, TQ induced a significant increase in the protein level of PTEN (*p* < 0.001) in comparison with untreated cells (Figure 5 and Table 2). 

## 3. Discussion

Despite the significant advances in the treatment of leukemia, the side effects of chemotherapy and the resistance to tyrosine kinase inhibitors (TKIs) represent the main challenges in treating leukemia patients [20,28]. Therefore, searching for new therapeutic alternatives could significantly impact leukemia treatment.

Constitutive activation of JAK/STAT and PI3K/Akt/mTOR signaling pathways are crucial in the pathogenesis of leukemia [16,22,29]. Therefore, proteins involved in JAK/STAT and PI3K/Akt/mTOR pathways provide vital targets for treating leukemia.

The anti-cancer activities of TQ have been demonstrated in several cancers [24,30]. TQ has been shown to inhibit many signaling pathways, including PI3K/Akt /mTOR and JAK/STAT signaling in several cancer cells [23,31]. However, the underlying mechanisms of TQ’s anti-leukemia activities have not been extensively studied. 

We have previously reported that TQ exerted anti-leukemia effects by inducing antiproliferative effects and enhancing apoptosis in MV4-11 AML cells and K562 CML cells and through hypomethylation of JAK/STAT and PI3K/Akt/mTOR-negative regulator genes in MV4-11 AML cells [27,32]. In the current study, we have investigated the effect of TQ on JAK/STAT and PI3K/Akt/mTOR signaling pathways in MV4-11 AML cells and K562 CML cells. 

The results of the present study showed that TQ induced growth inhibitory effect in K562 cells in a time and dose-dependent manner (Figure 1). Increasing TQ concentration induced a gradual increase in the growth inhibition up to 99% after 72 h of treatment. In addition, the growth inhibitory effect was increased significantly from 57–86% and 97% after incubation with 21 μM TQ for 24 h, 48 h, and 72 h, respectively, indicating a time-dependent course of TQ’s growth inhibitory effect. Our results are consistent with our previous findings of TQ’s time and dose-dependent course of growth inhibition in MV4-11 cells [27].

Regulation of PI3K/Akt/mTOR signaling has a crucial role in normal hematopoiesis [15]. Aberrant expression of *PI3K*, *Akt*, *mTOR,* and *PTEN* plays an essential role in the leukemogenesis [33,34]. Therefore, genes involved in PI3K/Akt/mTOR signaling provide critical molecular targets for the treatment of myeloid leukemia. In the current study, to further explore the underlying mechanism behind the antiproliferative and enhancement of apoptotic activities of TQ on MV4-11 and K562 cells, the effect of TQ on the expression of *PI3K*, *Akt*, *mTOR,* and *PTEN* was evaluated in MV4-11 and K562 cells. The findings revealed a significant decrease in the mRNA levels of PI3K, Akt, and mTOR in both MV4-11 cells (Figure 2A) and K562 cells (Figure 2B). Additionally, the findings showed that the mRNA level of PTEN was significantly increased in MV4-11 and K562 cells after TQ treatment (Figure 2A,B). The results of this study are in agreement with a recent study that showed that the in vivo administration of TQ downregulated the mRNA levels of PI3K, Akt, and mTOR in hamster oral cancer [35]. The results of this study are also in agreement with a previous study that reported that TQ treatment increased the expression of PTEN at the mRNA level in MCF-7 human breast adenocarcinoma cells [36], and Her2+ and Her2− breast cancer cells [37].

In AML cells, activation of FLT3 tyrosine kinase is frequent and results in dysregulation of PI3K/Akt/mTOR [16]. In addition, BCR-ABL oncogenic protein induces activation of PI3K/Akt/mTOR pathway by several mechanisms in CML cells [22,29]. Moreover, the inactivation of PTEN is associated with leukemogenesis and is frequently inhibited in AML and CML cells [18]. Thus, inhibition of PI3K/Akt/mTOR signaling and re-expression of PTEN protein might provide important therapeutic targets for the treatment of myeloid leukemia.

In the current study, the effect of TQ on the expression of PTEN protein and the activation status and the expression of proteins involved in PI3K/Akt/mTOR signaling were evaluated in MV4-11 and K562 cells. The findings indicated that TQ significantly reduced the phosphorylation of PI3K and Akt proteins in MV4-11 cells (Figure 4 and Table 1), and K562 cells (Figure 5 and Table 2). The results also showed that TQ dramatically reduced the protein expression level of PI3K and Akt, and increased the protein expression level of PTEN in MV4-11 cells (Figure 4 and Table 1) and K562 cells (Figure 5 and Table 2). These findings were supported by the results of gene expression analysis, indicating that TQ downregulated PI3K and Akt and upregulated PTEN at the mRNA and protein levels. Similarly, TQ reduced the phosphorylation and the protein expression level of Akt in ML-1, MV4-11, and Kasumi-1 leukemia cells [24], MDA-MB-231, and MDA-MB-436 triple-negative breast cancer (TNBC) cells [25], and Her2+ and Her2− breast cancer cells [37]. Moreover, TQ inhibited the phosphorylation of PI3K and Akt in oral squamous cell carcinoma (KB) cells [38]. Another study has demonstrated that the PTEN protein level was increased, and the PI3K protein level and the phosphorylation of its downstream molecules; Akt and mTOR, were decreased in K562 cells after treatment with ND-09 inhibitory compound [39]. 

Aberrant activation of JAK/STAT signaling has been associated with the development of hematopoietic malignancies including AML and CML [40,41]. FLT3-ITD and BCR-ABL oncoproteins have constitutive tyrosine kinase activities, which induce hyperactivation of signal transduction pathways, including JAK/STAT signaling [22,42]. Thus, inhibition of JAKs and STATs represents a potential strategy for treating AML and CML [43]. 

In this study, TQ’s effect on JAK/STAT signaling in MV4-11 cells was evaluated. The results showed that TQ significantly decreased the phosphorylation and protein levels of STAT3, STAT5, and JAK2 proteins in MV4-11 cells (Figure 3 and Table 1) and K562 cells (Data not shown) compared to untreated cells. 

The results of the current study indicated that TQ induced the inhibition of JAK/STAT signaling, which inhibited growth and enhanced apoptosis in MV4-11 cells as reported previously in our published data [27]. These findings were consistent with our previous findings in which TQ decreased the phosphorylation and the protein levels of JAK2, STAT3, and STAT5 in HL60 AML cells [44]. The results were also supported by the findings of a previous study in which TQ treatment reduced the phosphorylation of JAK2 and STAT3 in SK-MEL-28 melanoma cells [30]. Additionally, Pang et al. 2017 have reported that TQ treatment reduced the phosphorylation and protein expression level of STAT5 in ML-1, MV4-11, and Kasumi-1 leukemia cells [24]. Another study demonstrated that TQ treatment inhibited the phosphorylation of STAT3 in HGC27, SGC7901, and BGC823 gastric cancer cells, and inhibited the phosphorylation of JAK2 in HGC27 cells [14]. 

Taken together, TQ downregulates the expression of *PI3K*, *Akt*, and *mTOR,* and upregulates the expression of *PTEN*. In addition, TQ inhibits PI3K/Akt/mTOR and JAK/STAT signaling in myeloid leukemia cells, which was associated with growth inhibition of leukemia cells.

## 4. Materials and Methods

### 4.1. Cell Culture

*FLT3*-ITD positive MV4-11 AML cells were purchased from Elabscience Biotechnology Co., Ltd., (Wuhan, China). *BCR-ABL* positive K562 CML cells were purchased from the American Type Culture Collection (ATCC, Manassas, VA, USA). Roswell Park Memorial Institute (RPMI) medium, fetal bovine serum (FBS), and penicillin/streptomycin (P/S) solution were purchased from Elabscience Biotechnology Co., Ltd., (Wuhan, China). MV4-11 and K562 cells were cultured in an RPMI medium supplemented with 10% FBS and 1% P/S and maintained in a humidified incubator supplied with 5% carbon dioxide (CO_2_) at 37 °C. The cells were subcultured at a density of 5 × 10^4^ cells/mL until 80% confluency. 

### 4.2. Treatment with Thymoquinone 

Thymoquinone (>98% pure), purchased from Sigma-Aldrich (Sigma-Aldrich Corp., Louis, MO, USA). A 30 mM stock solution of TQ was prepared in 1 mL of 100% dimethyl sulfoxide (DMSO) and kept at −80 °C. To prevent cytotoxicity of DMSO, the stock solution of TQ was serially diluted with culture media. After that, treatment concentrations of TQ (15 µM for K562 cells treatment and 5.5 µM for MV4-11 cells treatment) were prepared with final concentrations of DMSO of 0.048% and 0.017% for K562 cells treatment and MV4-11 cells treatment, respectively. 

### 4.3. Cytotoxicity Analysis by WST-8 Assay

In the current study, the cytotoxic effect of TQ on K562 cells was investigated using the Cell counting water-soluble tetrazolium salt-8 (WST-8) kit (Nacalai Tesque, Inc., Kyoto, Japan). K562 cells were cultured in 96-well culture plates at a density of 5 × 10^3^ viable cells/100 μL RPMI medium. The plates were then incubated in a humidified incubator (5% CO_2_, at 37 °C) for 24 h. After that, TQ solutions were prepared at varying concentrations (0 μM, 3 μM, 6 μM, 9 μM, 12 μM, 15 μM, 18 μM, 21 μM, 24 μM, 27 μM, and 30 μM), and 10 μL were added to the desired wells. The plates were then incubated for 24 h, 48 h, and 72 h. Following treatment, 10 μL of WST-8 solution was added to each well and incubated for another 4 h. The relative cell viability was assessed by measuring the absorbance at 450 nm (reference wavelength 600 nm) using a microplate reader (Infinite M200, Tecan, Männedorf, Switzerland). 

In our previous study, the cytotoxic effect of TQ on MV4-11 cells was investigated, and the IC_50_ values of TQ on MV4-11 cells were determined after 24 h, 48 h, and 72 h of treatment [27].

### 4.4. RNA Extraction

MV4-11 and K562 cells were treated with the IC_50_ of TQ; 5.5 µM and 15 μM, respectively, for 48 h. Total cellular RNA was extracted from both TQ-treated and untreated cells using ReliaPrep™ RNA Cell Miniprep System RNA extraction kit (Promega, Madison, WI, USA) according to the instructions of the manufacturer. Nanodphotometer (Implen, Weslake Village, CA, USA) was used to measure the purity and concentration of the extracted RNA.

### 4.5. Reverse Transcriptase Quantitative PCR (RT-qPCR)

First, 100 ng of RNA was reverse transcribed into cDNA using GoTaq^®^ 2-Step RT-qPCR System kit (Promega, USA) according to the manufacturer’s instructions. The expression of *PI3K*, *Akt*, *mTOR,* and *PTEN* was analyzed using SYBR Green-based GoTaq 2-Step RT–qPCR System kit (Promega, Madison, WI, USA). The PCR reactions were performed in triplicates. Each PCR reaction contained 10 μL of GoTaq^®^ qPCR Master Mix (2×), 0.2 μL of CXR reference dye, 1 μL of forward (20×) and reverse primers (20X), 6.8 μL nuclease-free water, and 2 μL of cDNA template (2 μL of nuclease-free water were added to the no template control (NTC)). The cycling conditions for the RT-qPCR reactions were as follows: activation of GoTaq^®^ DNA Polymerase, 2 min at 95 °C; denaturation step, 40 cycles at 95 °C for 15 s; annealing and extension, 40 cycles at 60 °C for 1 min. RT-qPCR was analyzed using Applied Biosystem StepOnePlus ™ thermocycler (Applied Biosystems, Foster City, CA, USA). The sequences of the primer sets used for PCR amplification are listed in Table 3. Data were analyzed by StepOne Software v2.3 (ABI step one plus, Foster City, CA, USA). The relative expression levels of the genes were determined using RT-qPCR and the 2^−∆∆Cq^ method [45] after normalization to the endogenous reference *β-actin*, and the results were presented as fold changes.

### 4.6. Total Cell Lysates Preparation

The IC_50_ of TQ; 5.5 µM and 15 μM were used to treat MV4-11 and K562 cells, respectively, for 48 h. After that, the treated and untreated cells were lysed with RIPA buffer containing a protease inhibitor cocktail (Nacalai Tesque, Kyoto, Japan) and phosphatase inhibitor solution (Nacalai Tesque, Kyoto, Japan). The protein concentrations were measured by Bradford protein assay [49], using bovine serum albumin (BSA) and Coomassie brilliant blue (CBB) solution from Nacalai tesque, Inc. (Kyoto, Japan). 

### 4.7. Protein Expression Analysis Using JESS Simple Western Analysis

Protein expression was investigated using the capillary-based Protein Simple Jess system (Jess; Protein Simple, San Jose, CA, USA). Briefly, 1 mg/mL concentration of the cell lysates was prepared after dilution with a sample buffer. Each sample was mixed with a master mix (1× fluorescent standard, 1× sample buffer, and 40 mM dithiothreitol) and heated at 95 °C for 5 min to denature the samples. After that, 3 μL of the denatured proteins and 10 μL of each protein normalization solution, chemiluminescent substrate, primary antibodies, and HRP-conjugated anti-mouse secondary antibodies were pipetted into the appropriate wells of the assay plate. Biotinylated ladder cartridge (12–230 kDa) was integrated for each assay. Subsequently, the plate and the capillaries were subjected to Jess machine for automated protein electrophoresis, blocking, antibody incubation, and signal detection. Within-capillary total protein normalization was performed to account for any differences in protein loading. Normalization reagent allowed the detection of total proteins in the capillary through the binding to the amine group by a biomolecule while also removing housekeeping proteins that might cause inconsistency and unreliability in the protein’s expression. With the Jess technology, no loading control is required. Chemiluminescent reactions were analyzed by Compass software (ProteinSimple). Relative protein amounts were assessed using the corrected area of chemiluminescent peaks. The primary antibodies used for Jess simple western analysis and their suppliers are listed in Table 4.

## 5. Conclusions

The effect of TQ against MV4-11 AML cells and K562 CML cells was evaluated. Based on our current findings, it can be concluded that TQ-induced growth inhibition of leukemia cells was associated with reduced expression of *PI3K*, *Akt*, and *mTOR* genes and re-expression of *PTEN* gene at the mRNA and protein levels. Moreover, TQ induced inhibition of PI3K/Akt/mTOR and JAK/STAT signaling pathways, which was associated with inhibition of proliferation of the leukemia cells. The results of this study suggest that TQ has potential anti-leukemic effects on both AML and CML cells. However, further experiments are required to evaluate the molecular mechanisms underlying TQ’s anti-leukemia activities in vivo and in clinical trials.

## Figures and Tables

**Figure 1 pharmaceuticals-15-01123-f001:**
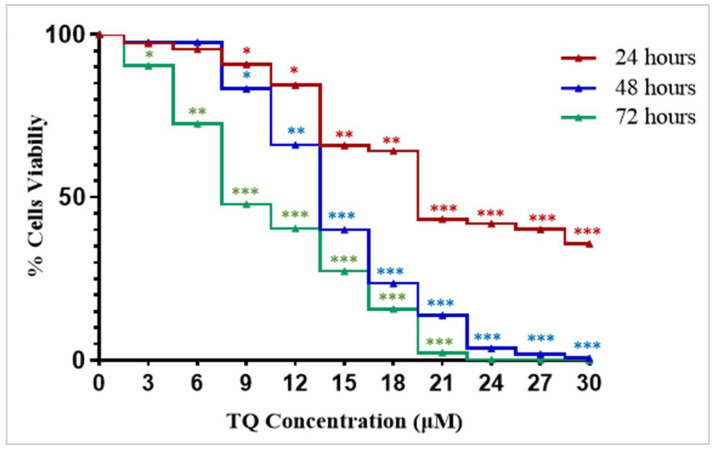
TQ’s dose and time-dependent inhibitory effects on the proliferation of K562 cells. The WST-8 assay was used to assess the cytotoxicity effects of TQ at different concentrations (3 μM, 6 μM, 9 μM, 12 μM, 15 μM, 18 μM, 21 μM, 24 μM, 27 μM, and 30 μM) on K562 cells after 24 h, 48 h, and 72 h of treatment, the IC_50_ values were 23 μM, 15 μM, and 11 μM, respectively. Wilcoxon signed ranks test was conducted and the results are stated as the median of four independent experiments, * *p* < 0.05, ** *p* < 0.01, and *** *p* < 0.001 are significant versus untreated control cells.

**Figure 2 pharmaceuticals-15-01123-f002:**
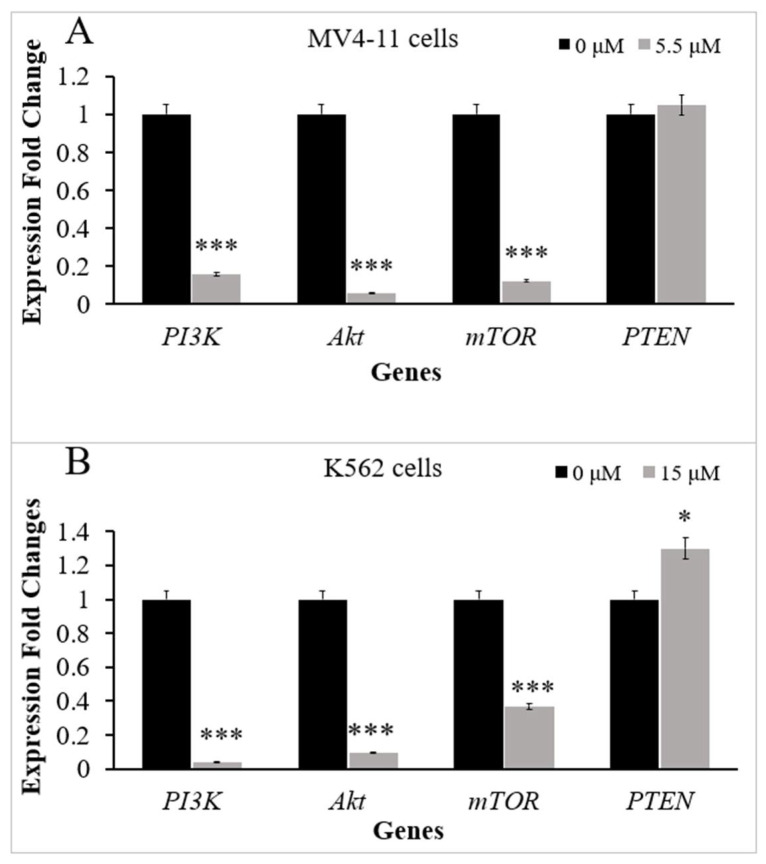
Effect of TQ on PI3K/Akt/mTOR signaling expression in MV4-11 and K562 cells. RT-qPCR analyses were performed. The graph demonstrates the downregulation of *PI3K, Akt,* and *mTOR* genes and upregulation of *PTEN* in (**A**) MV4-11 cells and (**B**) K562 cells after treatment with TQ. The data represent three separate experiments and are presented as the median and interquartile range (error bars). Wilcoxon signed ranks test was conducted. Where * *p* < 0.05 and *** *p* < 0.001 are significant versus untreated control cells.

**Figure 3 pharmaceuticals-15-01123-f003:**
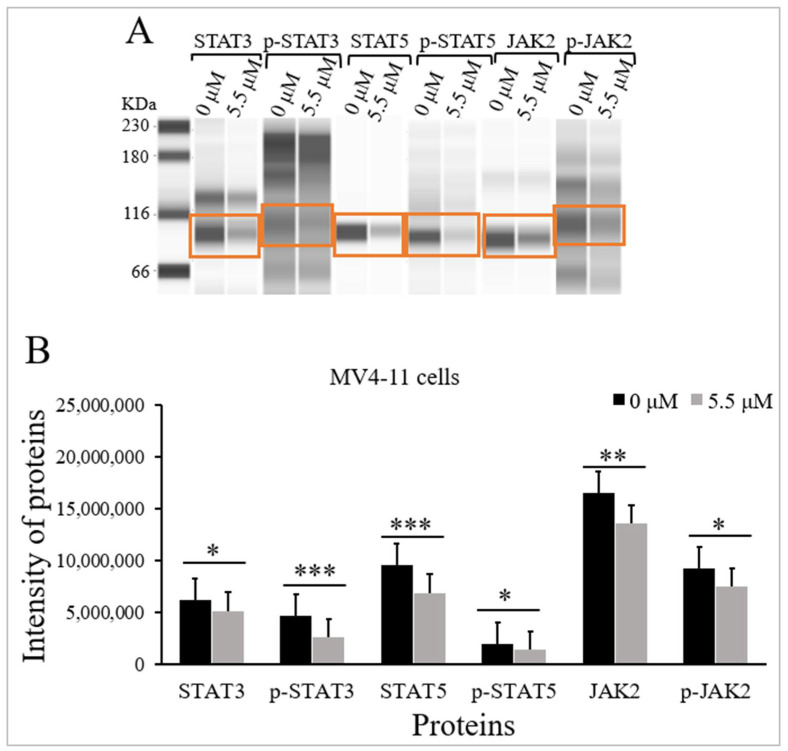
Effect of TQ on the activation status of JAK/STAT signaling in MV4-11 cells. The cells were incubated for 48 h with 5.5 μM TQ. The protein levels were determined by Jess simple western analysis. (**A**) Images of Jess simple western analysis for the target proteins before and after TQ treatment. (**B**) The bar graph demonstrates the reduction of STAT3, p-STAT3, STAT5, p-STAT5, JAK2, and p-JAK2 protein levels in MV4-11 cells after TQ treatment. Wilcoxon signed rank test was conducted, and the results are presented as medians and interquartile range (error bars) (n = 3), where * *p* < 0.05, ** *p* < 0.01, *** *p* < 0.001 are significant versus untreated control cells.

**Figure 4 pharmaceuticals-15-01123-f004:**
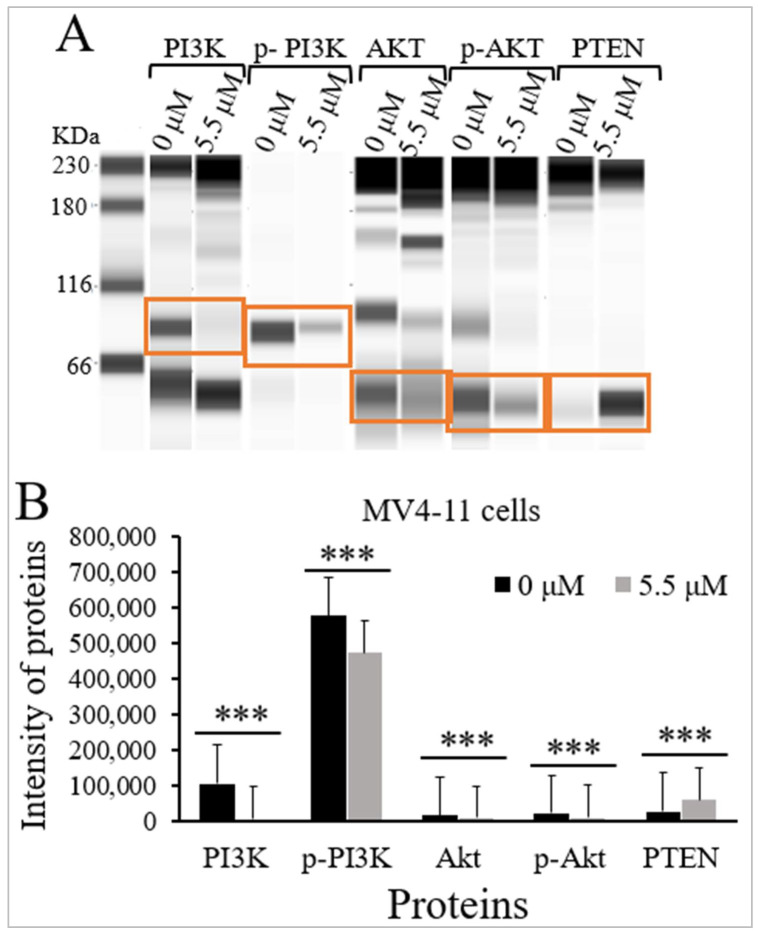
Effect of TQ on the activation of PI3K/Akt/mTOR signaling in MV4-11 cells. The cells were incubated for 48 h with 5.5 μM TQ. The protein levels were determined by Jess simple western analysis. (**A**) Images of Jess simple western analysis for the target proteins before and after TQ treatment. (**B**) The bar graph demonstrates the reduction of PI3K, p-PI3K, Akt, and p-Akt, and the increase of PTEN protein levels in MV4-11 cells after TQ treatment. Wilcoxon signed rank test was applied, and the results are stated as medians and interquartile range (error bars) (n = 3), *** *p* < 0.001 is significant versus untreated control cells.

**Figure 5 pharmaceuticals-15-01123-f005:**
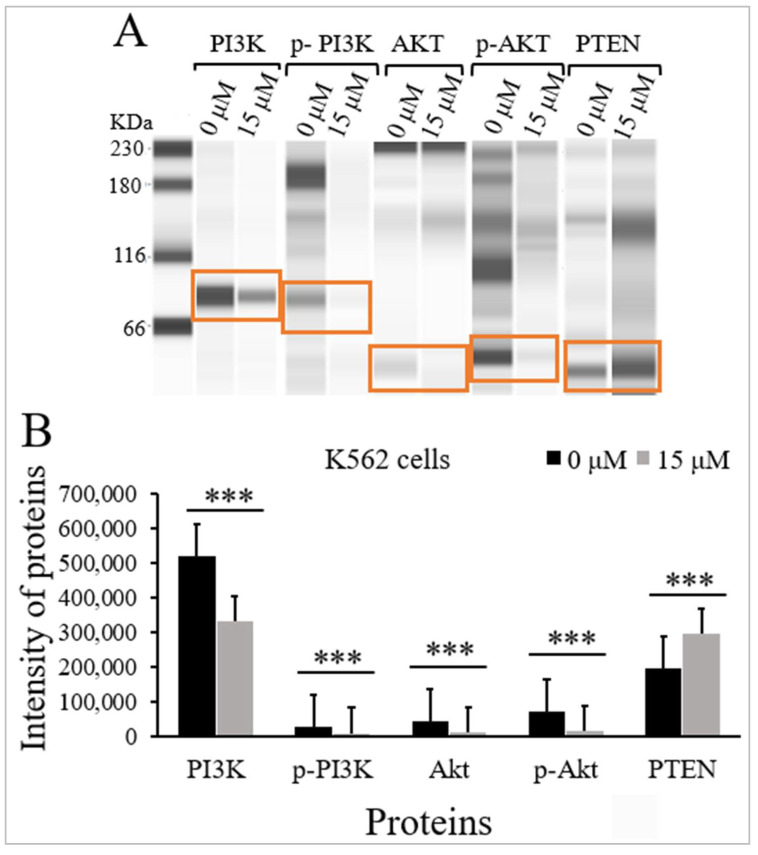
Effect of TQ on the activation of PI3K/Akt/mTOR signaling in K562 cells. The cells were incubated for 48 h with 15 μM TQ. The protein levels were determined by Jess simple western analysis. (**A**) Images of Jess simple western analysis for the target proteins before and after TQ treatment. (**B**) The bar graph demonstrates the reduction of PI3K, p-PI3K, Akt, and p-Akt, and the increase in PTEN protein levels in K562 cells after treatment with TQ. Wilcoxon signed rank test was conducted, and the results are presented as medians and interquartile range (error bars) (n = 3), *** *p* < 0.001 is significant versus untreated control cells.

**Table 1 pharmaceuticals-15-01123-t001:** Medians of protein intensity in MV4-11 cells.

Proteins	M.W (kDa)	Intensity of Proteins Median (IqR) (n = 3)		*p*-Value ^a^
		Untreated MV4-11 Cells	Treated MV4-11 Cells	
STAT3	80–90	6,238,528 (382,810)	5,156,854 (196,459)	<0.05
p-STAT3	95	4,644,236 (217,641)	2,588,108 (131,717)	<0.001
STAT5	91	9,545,158 (541,613)	6,908,133 (315,229)	<0.001
p-STAT5	92	1,944,235 (463,715)	1,376,497 (681,108)	<0.05
JAK2	130	16,523,707(831,602)	13,585,639 (719,317)	<0.01
p-JAK2	130	9,233,971 (691,001)	7,481,068 (592,419)	<0.05
PI3K	85	107,064 (45,613)	6627 (8731)	<0.001
p-PI3K	89	575,923 (104,621)	473,130 (85,372)	<0.001
Akt	62	17,443 (4794)	9897 (7803)	<0.001
p-Akt	60	22,879 (9117)	11,646 (2712)	<0.001
PTEN	60	28,705 (9062)	60,137 (11,823)	<0.001

^a^ Wilcoxon signed ranks test was applied. The median of protein expression is significant versus untreated control cells.

**Table 2 pharmaceuticals-15-01123-t002:** Medians of protein intensity in K562 cells.

Proteins	M.W (kDa)	Intensity of Proteins Median (IqR) (n = 3)		*p*-Value ^a^
		Untreated K562 Cells	TQ-Treated K562 Cells	
PI3K	85	520,529 (73,119)	332,532 (57,649)	<0.001
p-PI3K	89	29,977 (6881)	9633 (437)	<0.001
Akt	62	43,372 (9653)	12,165 (1711)	<0.001
p-Akt	60	72,651 (8091)	16,152 (5793)	<0.001
PTEN	60	194,916 (39,293)	295,205 (11,719)	<0.001

^a^ Wilcoxon signed ranks test was applied. The median of protein expression is significant versus untreated control cells.

**Table 3 pharmaceuticals-15-01123-t003:** The sequences of the primer set for RT-qPCR analysis.

Gene	Accession Number	Primer Set Sequences (5′–3′)	Reference
*PI3K*	NG_012113.2	Forward: TTAGCTATTCCCACGCAGGA Reverse: CACAATAGTGTCTGTGACTC	[46]
*AKT*	NG_012188.1	Forward: CTGAGATTGTGTCAGCCCTG Reverse: CACAGCCCGAAGTCTGTGATCTTA	[47]
*mTOR*	NG_033239.1	Forward: ATGCAGCTGTCCTGGTTCTC Reverse: AATCAGACAGGCACGAAG	[46]
*PTEN*	AF_067844.1	Forward: TGGATTCGACTTAGACT GACCT Reverse: TTTGGCGGTGTCATAATGTCTT	[33]
*β-actin*	NC_000071.7	Forward: CTGGCACCCAGGACAATG Reverse: GCCGATCCACACGGAGTA	[48]

**Table 4 pharmaceuticals-15-01123-t004:** List of primary antibodies used for Jess simple western analysis.

Primary Antibodies	Catalogue Number	COMPANY
Anti-JAK2	NBP2-59451	Novus Biologicals, LLC, Littleton, CO, USA
Anti-phospho-JAK2	NBP2-67429	Novus Biologicals, LLC, Littleton, CO, USA
Anti-STAT3	MAB1799	R&D System, Minneapolis, MN, USA
Anti-phospho-STAT3	AF4607	R&D System, Minneapolis, MN, USA
Anti-STAT5	AF2168	R&D System, Minneapolis, MN, USA
Anti-phospho-STAT5	MAB41901	R&D System, Minneapolis, MN, USA
Anti-PI3K	MAB2998	R&D System, Minneapolis, MN, USA
Anti-phospho-PI3K	PA5-177387	Invitrogen, Waltham, MA, USA
Anti-Akt	MAB2055	R&D System, Minneapolis, MN, USA
Anti-phospho-Akt	AF887	R&D System, Minneapolis, MN, USA
Anti-PTEN	AF847	R&D System, Minneapolis, MN, USA

## Data Availability

Data sharing contain in this article.

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
