# Peer review of "Thymoquinone Inhibits JAK/STAT and PI3K/Akt/ mTOR Signaling Pathways in MV4-11 and K562 Myeloid Leukemia Cells"

_pharmaceuticals, 2022, doi:10.3390/ph15091123_

Round 1

Reviewer 1 Report (Previous Reviewer 2)

Dear authors, thank you for taking into account my concerns. All of my prior points no longer apply because the molecular docking analysis has been deleted from the manuscript.

Author Response

Date: 31 /8/ 2022

Dear Editor

Thank you very much for giving us the opportunity to submit the revised manuscript entitled “Thymoquinone Inhibits JAK/STAT and PI3K/Akt/ mTOR Signaling Pathways in MV4-11 and K562 Myeloid Leukemia Cells” to the Pharmaceuticals journal.

We appreciate the time and effort that you and the reviewers have dedicated to providing valuable feedback on our manuscript. We have revised the manuscript as shown in the uploaded manuscript_track changes.

Reviewer 1

The manuscript can be accepted without further modification

Authors Response:

Dear reviewer, thank you very much for your recommendation. “The manuscript can be accepted without further modification”

Reviewer 2 Report (Previous Reviewer 4)

The manuscript can be accepted for publication without further modifications.

Author Response

Date: 31 /8/ 2022

Dear Editor

Thank you very much for allowing us to submit the revised manuscript entitled “Thymoquinone Inhibits JAK/STAT and PI3K/Akt/ mTOR Signaling Pathways in MV4-11 and K562 Myeloid Leukemia Cells” to the Pharmaceuticals journal.

We appreciate the time and effort you and the reviewers dedicated to providing valuable feedback on our manuscript. We have revised the manuscript as shown in the uploaded manuscript track changes.

Reviewer 2

Comment: Dear authors, thank you for taking into account my concerns. All my prior points no longer apply because the molecular docking analysis has been deleted from the manuscript.

Authors Response:

Dear reviewer, your comments were valuable and highly appreciated.

This manuscript is a resubmission of an earlier submission. The following is a list of the peer review reports and author responses from that submission.

Round 1

Reviewer 1 Report

Authors have revised the manuscript properly.

Reviewer 2 Report

The authors report the effect of thymoquinone on various signaling pathways linked to leukemia. Thymoquione was first described in 1963 and since then was extremely well studied. It has been demonstrated numerous times that it may have anticancer properties, including the ability to deregulate MAPK and AKT/PKB pathways [10.1371/journal.pone.0023741, 10.1371/journal.pone.0046641]. On the other hand, thymoquinone has a high redox activity and is a well-known PAIN compound [10.1021/acs.jnatprod.5b00947]. Though its detailed mode of action is yet unknown, thymoquione's activity is frequently associated with redox cycling or acting as a Michael acceptor.

The authors are addressing this question in this manuscript and discuss a potential mechanism of action among other findings (I have no concerns about generated biology data). This is what I consider to be the most problematic aspect of the manuscript.

The primary method chosen to substantiate their assertions was molecular docking. I do not think this is a good strategy. First of all, evidence from experiments should be used to support such claims (direct binding in the specified binding sites): X-ray or at least protein level testing.

Additionally, according to all of the binding models that have been presented, tymoquione is predicted to bind in a non-covalent manner, which is unexpected given the structure of the compound and the presence of nucleophilic amino acids in the binding sites. So, I consider this section of the manuscript to be totally speculative, especially taking into account the low predicted values of the scoring function (misleadingly referred to as "binding energy"). Although the authors acknowledge that molecular docking is frequently employed to uncover novel inhibitors of tyrosine kinases, they use it here as a post-hoc approach to support their claims. 

I think this section of the manuscript is unworthy of publication, hence the entire work should be rewritten and is not publishable in its current state. 

Reviewer 3 Report

The manuscript “Thymoquinone Inhibits JAK/STAT and PI3K/Akt/ mTOR Signaling Pathways in Myeloid Leukaemia Cells: Protein Analysis and Molecular Docking Study” aimed fully understand the effect mechanism of Thymoquinone (TQ), which is one of the main 18 constituents of Nigella sativa, on leukemia. Previous reports have shown anti-cancer activities of TQ in several cancers.

The authors investigated the effect of TQ on JAK/STAT and PI3K/Akt/mTOR signaling pathways in MV4-11 AML cells and K562 CML cells. Their docking studies identified TQ as a potential inhibitor of the enzymatic activities of PI3K, Akt, mTOR, FLT3-ITD, and BCR-ABL tyrosine kinases. Additionally, the results showed that TQ induced growth inhibitory effect in K562 cells in a time and dose-dependent manner.

The manuscript was well designed; the experiments were adequately carried out, presented, and discussed. The text, in general, is well presented. I recommend its publication after some modifications. 

The authors could provide some literature data about molecular docking of interactions between natural compounds and studied enzymatic binding sites (for instance, in line 421, page 15). It would be an interesting comparison source for readers, which would high the citation index of the article.

Reviewer 4 Report

My comments are reported in attached pdf file.
